# Silicate Inhibits the Cytosolic Influx of Chloride in Protoplasts of Wheat and Affects the Chloride Transporters, *TaCLC1* and *TaNPF2.4/2.5*

**DOI:** 10.3390/plants11091162

**Published:** 2022-04-26

**Authors:** Albert Premkumar, Muhammad Tariq Javed, Katharina Pawlowski, Sylvia M. Lindberg

**Affiliations:** 1Bharathiyar Group of Institutes, Guduvanchery 603202, India; albertprem@gmail.com; 2Department of Botany, Faculty of Life Sciences, Government College University, Faisalabad 38000, Pakistan; mtariqjaved@gcuf.edu.pk; 3Department of Ecology, Environment and Plant Sciences, Stockholm University, SE-11418 Stockholm, Sweden; katharina.pawlowski@su.se

**Keywords:** chloride, cytosolic uptake, silicate, *TaCLC1*, *TaNPF 2.4/2.5*, wheat

## Abstract

Chloride is an essential nutrient for plants, but high concentrations can be harmful. Silicon ameliorates both abiotic and biotic stresses in plants, but it is unknown if it can prevent cellular increase of chloride. Therefore, we investigated the influx of Cl^−^ ions in two wheat cultivars different in salt sensitivity, by epifluorescence microscopy and a highly Cl^−^-sensitive dye, MQAE, N-[ethoxycarbonylmethyl]-6-methoxy-quinolinium bromide, in absence and presence of potassium silicate, K_2_SiO_3_. The Cl^−^-influx was higher in the salt-sensitive cv. Vinjett, than in the salt-tolerant cv. S-24, and silicate pre-treatment of protoplasts inhibited the Cl^−^-influx in both cultivars, but more in the sensitive cv. Vinjett. To investigate if the Cl^−^-transporters *TaCLC1* and *TaNPF2.4/2.5* are affected by silicate, expression analyses by RT-qPCR were undertaken of *TaCLC1* and *TaNPF 2.4/2.5* transcripts in the absence and presence of 100 mM NaCl, with and without the presence of K_2_SiO_3_. The results show that both transporter genes were expressed in roots and shoots of wheat seedlings, but their expressions were differently affected by silicate. The *TaNPF2.4/2.5* expression in leaves was markedly depressed by silicate. These findings demonstrate that less chloride accumulates in the cytosol of leaf mesophyll by Si treatment and increases salt tolerance.

## 1. Introduction

### 1.1. Salinity Stress in Plants

Salinity stress in plants, mainly caused by both Na^+^ and Cl^−^ ions, is a challenge for cultivation of agricultural crops, as it has negative impacts on plant growth and yield at high concentration. However, chloride ions at a low concentration (50–100 µM) in the root medium are essential micronutrients for higher plants [1]. Besides as a nutrition, Cl^−^ has many other functions in plants, for instance in the PSII electron transport chain of photosynthesis, in turgor- and osmoregulation, as activator of enzymes, and as a regulator of cellular pH [2,3,4,5,6]. In halophytes and some glycophytes Cl^−^ even acts as a macronutrient by increasing the plants’ fresh and dry weights, and leaf expansion [7,8]. Halophytes can survive at 100–200 mM Na^+^ and Cl^−^ in the cytosol, but the tissue concentration can be much higher, >500 mM [7].

Compared with sodium, Cl^−^ is less investigated and has been underestimated as a toxic element [9]. Experiments with faba bean showed that high Cl^−^ could be as toxic as high Na^+^ [10]. A reduced uptake of chloride into root xylem was correlated with salinity tolerance in wheat and rice [11,12].

To prevent chloride toxicity in the cell cytosol, and chloride accumulation in the shoot, plants can increase the efflux of chloride from the cell [13] or prevent loading into the xylem [14]. The transport is either active or passive depending on Cl^−^ concentration [3].

Cl^−^ can be taken up into root cells by Cl^−^/2H^+^ symport and passively by channels in non-saline conditions. A voltage-dependent anion channel in mesophyll cells, activated by rise of cytosolic calcium, was reported to mediate efflux of chloride ions depending on the very negative membrane potential, resulting in an outward-directed anion gradient [3,15,16]. Newer investigations have focused on identification and properties of different chloride transporters and channels in different plant species.

### 1.2. CLCs Channels Involved in Chloride Transport

Different plants have different numbers of CLCs; in wheat (*Triticum aestivum* L.) 23 *TaCLC* gene sequences were identified. The expression pattern of *TaCLC* family members was determined to be inducible under low nitrogen stress and salt stress [17]. In soybean under salt stress the GmCLC1 transporter is regulating chloride accumulation. Overexpression of the soybean transporter, GmCLC1, in *Arabidopsis thaliana* led to the sequestration of more Cl^−^ in the root and to less transport of Cl^−^ to the shoot, and thus, increased salt tolerance [18,19], suggesting that GmCLC1 probably is a Cl^−^/H^+^ antiporter as it is pH sensitive. This antiporter may be located in the tonoplast [20]. A recent report shows that in upland cotton the GhCLCg-1 chloride channel in the tonoplast is homologous to *Arabidopsis* AtCLCg and contributes to salt tolerance [21]. High external chloride concentrations increased the expression level of *GhCLCg-1* and decreased the Na^+^/K^+^ concentration ratio in roots, stem and leaves.

### 1.3. NPF Transporters Involved in Chloride Transport

For Cl^−^ efflux from the root, a member of the nitrate 1/peptide transporter family (NRT1/PTR/HFP), *NPF2.5*, was reported to be expressed in the plasma membrane of root cortical cells in *Arabidopsis*, and its expression was up-regulated by NaCl [22]. Also, another nitrate transporter, NPF2.4 was identified in *Arabidopsis* [20]. The corresponding gene was expressed in the stele and was downregulated by NaCl. The NPF2.4 transporter catalyzes passive efflux of Cl^−^ and was more permeable to Cl^−^ than to NO_3_^−^. Accumulation of Cl^−^ in the shoots was decreased in a micro-RNA knockdown but increased in overexpressors of *NPF2.4*. Therefore, this transporter is suggested to participate in long-distance transport of Cl^−^ from root to shoot in plants.

Recent findings suggest that Cl^−^ transport in maize is mediated by genes encoding the ZmNPF6.2 and ZmNPF6.6 proteins [1]. In *Arabidopsis*, xylem-driven Cl^−^-transport from root to shoot also was demonstrated via two S-type anion channels, AtSLAH3 and AtSLAH1 [23]. A large number of *NPF* genes (51 in *Capsella rubella*, 53 in *Arabidopsis*, 93 in rice, 139 in apple, 331 in wheat) have been identified, however, only a few of them have been functionally characterized [24,25].

### 1.4. NPF Transporters in Wheat (Triticum aestivum L.)

In wheat 16 *NPF* genes were found to be homologous to the ones in *Arabidopsis*, but little is known about their function; a TaNPF2.4 transporter has been identified [26]. In bread wheat a locus for Cl^−^ accumulation was suggested on chromosome 5A [27].

### 1.5. Silicon Effects

Silicon (Si), a non-essential element, is well-known for its beneficial effects on both quantity and quality of plant growth, development and yield of several plants. Application of Si to roots or by foliar spray or seed priming seems to be an eco-friendly strategy and silicon-supplementation has been shown to ameliorate the adverse consequences of both abiotic and biotic stresses [28]. The potential effect of salinity on morphological, physiological, biochemical and genetic parameters under salt stress was reported to be mitigated by Si [29,30,31]. The reduction of negative impact of salt stress by Si has been demonstrated in various plants and involves various mechanisms, such as increased water-use efficiency through osmoregulation, enhanced photosynthesis, better nutrient management by reduced sodium uptake together with an efficient regulation of the uptake of essential elements, and proper redox-homeostasis equilibrium by increased antioxidative capacity [32,33,34]. Although diverse protective effects of Si against salt stress were proposed, it is still unknown how Si affects chloride uptake in plant cells and if Si in some way influences the chloride transporters *TaNPF2.4/2.5* and *TaCLC1* in wheat.

### 1.6. Chloride in the Cytosol of Plant Cells

Usually the Cl^−^ concentration in the cytosol is only half of the Na^+^ concentration. In glycophytes under high salinity, chloride is toxic and causes a reduction in growth and yield, as it inhibits the net photosynthesis and metabolism in the cytosol and can diminish uptake of nitrate and phosphate by competition with chloride at the uptake sites [35].

One aim of this study was to investigate the cellular influx of chloride in the cytosol of the moderate-sensitive wheat cultivar Vinjett and compare this with the influx in the tolerant cv. S-24 upon NaCl addition; furthermore, we wanted to analyse how the direct addition of Si to protoplasts affected the influx of Cl^−^. Another aim was to gain knowledge on which ion transporters are involved in Cl^−^ transport in wheat cv. Vinjett and if the expression of these transporters is influenced by silicon. We focused on the *TaCLC1* and *TaNPF2.4/2.5* genes encoding proteins that are supposed to sequester chloride in vacuoles or to restrict the transport of chloride from root to shoot, respectively. We also analyzed the expression profiles of both *TaNPF2.4* and *TaNPF2.5* under salinity stress in the presence vs. absence of Si.

## 2. Results

### 2.1. Cytosolic Uptake of Chloride in Mesophyll Protoplasts

The Cl^−^-sensitive fluorescent compound, 6-methoxy-1-(3-sulfonatopropyl) quinolinium, SPQ, whose fluorescence can be quenched by Cl^−^, was earlier used to measure chloride uptake in tonoplast vesicles [36] and was independent of pH. We instead used a similar chloride sensitive dye MQAE, N-[ethoxycarbonylmethyl]-6-methoxy-quinolinium bromide, developed by Verkman et al. [37] to measure changes in protoplast fluorescence upon addition of NaCl in absence vs. presence of Si. This dye has greater sensitivity to Cl^−^ and a higher fluorescence quantum yield than SPQ. Although it is used for one-wavelength excitation measurements, and thus, cannot be used for strict concentration quantification, it can yield valuable information on chloride concentration changes. The results showed that the fluorescence decreased in % upon NaCl addition, which means that Cl^−^ concentration increased by influx (Figure 1).

The reaction was more pronounced in the moderately salt-sensitive cv. Vinjett than in the tolerant cv. S-24 at all treatments (Table 1).

When 100 mM NaCl was added to protoplasts without any pre-treatment with Si, a more pronounced increase in Cl^−^ influx was seen in both cultivars (50.28 % in cv. Vinjett and 46.71 % in cv. S-24) compared to protoplasts pre-treated with Si (45.65 % in cv. Vinjett and 43.30% in cv. S-24). Similarly, when 50 mM NaCl was added before Si, a larger increase in Cl^−^ influx (35.97 % in cv. Vinjett and 33.89 % in S-24) was observed than when Si was added before NaCl (30.33 % in cv. Vinjett and 26.82 % in cv.S-24). Thus, addition of Si before NaCl prevented the influx of chloride more than if NaCl was added first. Addition of Si before NaCl inhibited the chloride influx in a similar way in the two cultivars, but addition of Si after NaCl had little effect on the salt-tolerant cv. S-24.

### 2.2. RT-qPCR Was Used for Expression Analyses of TaCLC1 and TaNPF2.4 Genes

The expression analyses of the chloride channel gene *TaCLC1* showed that in control plants, the gene was expressed at higher levels in the roots than in the shoots (Figure 2). Both in roots and shoots, treatments with 100 mM NaCl, or with 1 mM Si, led to lower expression levels. However, a combined treatment with Si and NaCl increased the expression levels in both roots and shoots; in shoots expression levels became even higher than in control leaves.

In wheat *TaNPF2.5* shares 83.2% sequence similarity at the amino acid level with *TaNPF2.4*, and represents its closest homolog. Therefore, we used primer pairs previously published by Buchner and Hawkesford [26] for real-time PCR that were designed to amplify the transcripts of both isoforms. Expression analysis of *TaNPF2.4/2.5* showed a different pattern (Figure 3). The expression of *TaNPF2.4/2.5* was somewhat higher in roots than in shoots, but otherwise very similar in both organs. The presence of 100 mM NaCl did not change it. On the other hand, Si treatment lowered the expression in the shoots but increased it in the roots. Combined treatment with NaCl and Si reduced the expression levels in both roots and shoots.

## 3. Discussion

In plants exposed to salinity, uptake of Na^+^ is usually accompanied with the uptake of Cl^−^, loss of K^+^ and a transient cytosolic increase of Ca^2+^ [10,38,39,40]. Less focus has been devoted to uptake of Cl^−^ into the cytosol and its further translocation within the plant.

To our knowledge, this is the first time that the chloride sensitive dye MQAE is used to monitor Cl^−^ concentration changes in plant cells. This dye was used before in animal cells [41,42]. The MQAE labelled mesophyll protoplasts extracted from wheat cultivars showed a definite response to NaCl and Si additions (Appendix A).

When 50–100 mM NaCl were added to the protoplasts, the sensitive cv. Vinjett showed a higher degree of Cl^−^ uptake than the tolerant cv. S-24. The addition of Si reduced the Cl^−^ uptake. As a result, adding Si before NaCl led to a stronger inhibition of Cl^−^ uptake. This may be explained by the fact that Si affects the influx both at the plasma membrane as well as at the tonoplast of the protoplasts, as Si can easily pass the plasma membrane.

Our results with wheat cv. Vinjett showed that the two identified Cl^−^-transporter genes investigated, *TaCLC1* and *TaNPF2.4/2.5* were expressed in both roots and shoots. Theoretically, Si could affect the expression of both *TaNPF2.4/2.5* and *TaCLC1*.

CLC-type channels can have different locations within plant cells [19]. Experiments with upland rice indicated that the GhCLCg-1 channel is located in the vacuolar membrane and functions as a transport channel for Cl^−^ into the vacuoles [21]. Similar to the GsCLC-c2 channel in soybean, this leads to a diminished translocation of Cl^−^ to the shoots and thus to increased tolerance to salt stress [43].

Contrary to the general observations on soybean, quinoa, and *Arabidopsis thaliana*, the wheat cv. Vinjett showed significant downregulation of *TaCLC1* in both roots and shoots in response to 4-days salinity stress (100 mM NaCl). This could be due to the more pronounced inhibitory effect of 100 mM NaCl on cv. Vinjett, which is a salt-sensitive cultivar [44] and showed reduced root growth under these conditions (Appendix A). Si applied alone, or along with salinity led to a complex transcriptional pattern of *TaCLC1* in roots and shoots of cv. Vinjett. Our study confirms that Si applied to this wheat cultivar significantly reduced Cl^−^ uptake suggesting a control of Cl^−^ transport from roots to shoots. To obtain more insight into the expression patterns and functionality of the Cl^−^ transporter genes, more trials involving shifting the order of treatments, such as imposing Si to already salinity-stressed wheat cultivars, are required.

In wheat, *TaCLC1* was expressed at higher levels in roots than in shoots; therefore, it is likely that due to its action, more Cl^−^ is sequestered in the vacuoles of root cells and less is translocated to the shoots.

In our protoplast study, when 50 or 100 mM NaCl were added, Cl^−^ was taken up into the cytosol of leaf mesophyll protoplasts. This uptake was significantly diminished if Si was added before NaCl. This result corroborates the results of the expression analysis of *TaCLC1*, which showed that when seedlings are treated with Si and NaCl, compared with only NaCl, *TaCLC1* expression increases; this should lead to enhanced sequestration of Cl^−^ in the vacuoles, especially in shoots (Figure 2). It is suggested that Si addition thus increases the plant’s tolerance to high Cl^−^ concentrations. High chloride levels in the cytosol are toxic to most plants, but vacuolar chloride is not toxic [45].

The nitrate and chloride transporter NPF2.4 has been characterized both in *Arabidopsis* [20] and wheat [26,46]. It functions in the translocation of Cl^−^ from root to shoot, and the corresponding gene is expressed in the stele. Salinity stress did not affect the expression levels of *TaNPF2.4/2.5* in cv. Vinjett after 4-d treatment with 100 mM NaCl of one-week-old wheat seedling (Figure 3). Similar to our results, Wang et al., (2020) [25] also observed such differential expression and regulation in the expression patterns of wheat *TaNPFs* under various abiotic stress conditions.

Interestingly, in absence of salt stress Si treatment induced *TaNPF2.4/2.5* expression in roots of cv. Vinjett, while Vinjett shoots showed a marked reduction in expression. When comparing the expression profiles of both *TaCLC1* and *TaNPF2.4/2.5*, it is possible that in cv. Vinjett, the role of *TaCLC1* in Cl^−^ sequestration is taken over by *TaNPF2.4/2.5* in response to Si treatments at least in roots. Further experiments using knock-out mutants of both genes will offer additional insights on Si influences on Cl^−^ transport mechanisms. Salinity imposed separately, or in combination with Si (Figure 3), had a more noticeable inhibitory effect on *TaNPF2.4/2.5* transcription than found in *Arabidopsis*, as evidenced by the drastic reduction in *TaNPF2.4/2.5* transcript abundance in the combined treatments [20].

Under stress, *NPF* gene expression can be differentially regulated depending on the type of stress and duration, revealing a complex regulatory network. In wheat, the expression of *TaNPFs* was tissue-specific [20,46]. *TaNPF6.3* expression appeared to be abundant in both roots and shoots, whereas transcripts of *TaNPF6.1* and *TaNPF6.2* were found in abundance in roots, but not in shoots. As the Cl^−^ transport activity of *TaNPF2.4/2.5* is primarily interconnected with the transport of other ions such as NO_3_^−^, K^+,^ and H^+^ [20], simultaneous measurements of all these ions together with *TaNPF2.4/2.5* expression patterns are required to explore the mechanistic basis of the observed regulatory patterns.

In Si-treated plants the expression of *TaNPF2.4/2.5* increased in roots but decreased in shoots, which should have a positive effect on plant growth, as less Cl^−^ should be translocated to the shoots. Thus, our results show that Si has positive effects both on root and leaves of wheat seedlings and increases tolerance to high Cl^−^ concentration. The reduced concentration of Cl^−^ in the cytosol of leaf mesophyll cells in response to Si treatment may depend on Si effects both on *TaCLC1* and *TaNPF2.4/2.5*, but other Cl^−^ transporters may be involved as well.

## 4. Materials and Methods

### 4.1. Plant Material and Cultivation

Two wheat cultivars (*Triticum aestivum* L.), cv. Vinjett (Svalöf-Weibull, Malmö, Sweden) and cv. S-24 (Ayub Agricultural Research Institute Faisalabad, Punjab, Pakistan) were used for experiments with the cellular uptake of chloride. The seeds were treated with 10% chlorine solution for surface sterilization and then rinsed with distilled water 5–6 times. Then the seeds were soaked for 3 h in a CaSO_4_ solution (5 mM) and rinsed with water 5 times. The seeds were germinated under dark conditions in 1-L beakers and were placed on a Mira cloth (LIC, Stockholm, Sweden) covering a metal net and with a complete nutrient solution [2 mM KNO_3_, 1 mM Ca (NO_3_)_2_, 1 mM MgSO_4_, 1 mM KH_2_PO_4_, 0.5 mM Na_2_HPO_4_, 2.5 μM H_3_BO_3_, 0.3 μM CuSO_4_, 0.5 μM ZnSO_4_, 2 μM MnSO_4_, 0.01 μM (NH_4_)_6_Mo_7_O_24_ and 200 μM Fe-EDTA] [47]. The beakers were covered with thin transparent polythene sheets and placed in a growth chamber equipped with 400 W HQI-BT lamps (Osram, Munich, Germany). The temperature was maintained at 21 ± 1 °C with 12 h photoperiod at an irradiance of 200 µmol m^−2^ s^−1^ and relative humidity about 60%. Seedlings were grown with and without 1 mM K_2_SiO_3_ for separate experiments.

For the RT-qPCR investigation the seedlings were germinated in the same way. When the roots were developed, seedlings were treated in different ways for 4 d: T1, control with nutrient solution; T2, nutrient solution + 100 mM NaCl; T3, nutrient solution + 100 mM NaCl; and T4, nutrient solution + 100 mM NaCl + 1 mM K_2_SO_3_; and cultivated in a growth chamber under the same conditions as described above, but at an irradiance of 100 µmol m^−2^ s^−1^.

### 4.2. Protoplast Isolation and Loading with MQAE Dye

Leaf protoplasts of both wheat cultivars were isolated following the method described previously by Morgan et al. [44]. For cytosolic chloride determination, the protoplasts were loaded with the fluorescent anion indicator N-(6-methoxy-quinolyl) acetoxy ester (MQAE) that was developed by Verkman et al. [37]. The protoplasts were incubated with 5 mM dye concentration in a loading medium, composed of sorbitol (0.5 M), CaCl_2_ (0.1 mM), polyvinyl pyrrolidone (0.2% *w*/*v*) (PVP; Sigma), TRIS (5 mM) and MES (5 mM) at pH 5.5 (Medium A). Dye loading was carried out for 45 min at 37 °C under darkness.

After loading, the samples were transferred to a similar buffer as during the loading, but containing 1.0 mM CaCl_2_. The suspension pH was changed to 7.0 by use of 5 mM HEPES/TRIS buffer.

### 4.3. Fluorescence Measurement

Before measurement the protoplast suspension was kept in darkness at room temperature for 30 min to stabilize uniformly. For fluorescence measurement an epi-fluorescence microscope (Axiovert 10, Zeiss, Oberkochem, Germany), combined with a microprocessor, electromagnetic filter exchanger, photometer and personal computer was used. Measurements were performed with a Planneofluar × 40/0.75 objective (Zeiss, Oberkochem, Germany) for phase contrast. Only properly cytosolic-loaded protoplasts of similar size were used. They were laid on micro-slides covered with poly-L-lysine to attach them to the slides [44]. The excitation wavelength was 350 ± 10 nm and emission 460 ± 10 nm. In order to check the effect of different treatments on chloride influx, measurements were made with single protoplast before, and after, the addition of K_2_SiO_3_ (1 mM) and NaCl (50 or 100 mM) to the protoplast suspension. The fluorescence intensity associated with dye leakage to the solution was subtracted from the data prior to final calculations of in situ MQAE fluorescence.

Calibration of MQAE fluorescence was performed in situ on single protoplasts in the same buffer solution at pH 7.0 as used for measurements, but at different added NaCl concentrations as described by Verkman et al. [37]. A Stern-Volmer plot, F_0_/F_Cl_-1 potted against chloride concentration, shows a straight line for chloride concentrations between 20 and 100 mM (Appendix A).

### 4.4. Statistical Analysis

Each plot is a copy of printer plots and shows representative traces of a specific experiment repeated more than ten times with protoplasts from independent cultivations. Each value is the average of around 50 fluorescence determinations. Table 1 shows the results from measurements that were repeated up to ten times using protoplasts from independent cultivations. Statistical analysis of data was performed with analysis of variance (ANOVA) by using a statistical program Co-Stat version 6.2, Cohorts Software, 2003, Monterey, CA, USA. Significant differences between the treatments were determined by the one-way ANOVA and followed by a multiple comparison procedure with Tukey’s honestly significant difference method (at least *p* < 0.05).

### 4.5. Expression Analysis of TaCLC1 and Ta NPF2.4 in Roots and Leaves by RT-qPCR

#### 4.5.1. Total RNA Extraction

Root and leaf tissues were collected from the control and treated wheat plants and immediately flash-frozen in liquid nitrogen for further analysis. The total RNA was extracted using a Spectrum Plant Total RNA Kit (Sigma, Burlington, MA, USA) following the manufacturer’s instructions. To avoid DNA contamination, on-column DNase digestion (DNASE10/70) procedure was followed. The total RNA concentration and purity were determined using a NanoDrop ONE Spectrophotometer (Thermo Scientific, Waltham, MA, USA). Genomic DNA contamination was subsequently confirmed by PCR (DreamTaq TM Green PCR, Thermo Fisher Scientific Baltics UAB, Vilnius, Lithuania), using primers specific for a housekeeping gene (*Ta54227*). RNA samples with traces of gDNA contaminants were further digested using Heat&Run gDNA removal kit (ArcticZymes Technologies ASA, Tromsø, Norway).

#### 4.5.2. cDNA Synthesis

The RNA samples with absorption ratios of A260/A280 = 1.9–2.1 and A260/A230 ≈ 2.0 were used for cDNA synthesis. An aliquot of 1 μg of total RNA was used for cDNA synthesis with a final volume of 20 μL using a TATAA GrandScript cDNA synthesis kit (TATAA Biocenter AB, Gothenburg, Sweden) following the manufacturer’s instructions.

#### 4.5.3. Real-Time Quantitative (RT-qPCR) Analysis

For RT-qPCR, we used primers (Table 2) described in previous publications and all the primers showed a single peak map, no heteropeak, and also no peak in the negative control.

The standard curves with the log value on the x-axis and the Ct value on the y-axis to obtain the slope (K) and correlation coefficient (R_2_) with the following formula: E = [5 (1/−K) − 1] × 100% were generated by performing RT-qPCR on each gene using serially diluted cDNA samples. The amplification efficiency was between 95 and 120% with R2 > 0.99 correlation coefficient. The RNase-free water was used as a negative control template (NTC—no template control) to detect reagents or contamination during the experiment.

RT-qPCR reactions were performed in 96-well plates using the LightCycler^®^ 480 Real-Time System (Roche Diagnostics International AG, Rotkreuz, Switzerland or Roche Diagnostics, Mannheim, Germany) with Maxima SYBR Green qPCR Master Mix (Thermo Fisher Scientific Baltics UAB, Vilnius, Lithuania) according to manufacturer’s protocols in a 10-μL reaction. All reactions were performed in three technical replicates for each of three independent biological replicates.

A two-step amplification protocol was applied, with an initial denaturation cycle at 95 °C for 1o minutes, 45 cycles of denaturation at 95 °C for 15 s and annealing at 60 °C for 60 s; a melt analysis was then performed from 60 °C to 95 °C at 0.1 °C/s to verify the specificity of the desired amplicon. The expression of all analyzed genes was determined in each reaction using the threshold cycle (Ct value) and the Ct value was set automatically by the LightCycler^®^480 Software 1.5.0. SP3 software (version 1.5.0.39).

The raw data were analyzed to calculate the relative gene expression with normalization to three reference genes (*Ta2291*, *Ta2776*, and *Ta54227*) over multiple plates. Relative expression levels were calculated by normalizing the data to the geometric mean of three reference genes [51]. Relative fold change was calculated using the 2^−ΔΔCt^ method [52] and represented in the form of relative fold change in expression of genes under stress conditions as compared with its expression under control conditions using the PCR efficiencies and the Ct values

## 5. Conclusions

In this investigation. we employed the fluorescent Cl^−^-indicator MQAE to successfully monitor the cytosolic influx of chloride into leaf mesophyll protoplasts of two wheat cultivars during salt stress. Silicate (Si) administration before NaCl considerably reduced Cl^−^ absorption compared to Si addition after NaCl. Salt stressed wheat seedlings supplemented with Si revealed increased expression of *TaCLC1* in shoots, suggesting that it transports Cl^−^ into vacuoles. This clearly demonstrates that Si application boosts the plant’s tolerance to high Cl^−^ concentrations. Only the roots of cv. Vinjett showed a significant increase in *TaNPF2.4/2.5* expression in response to Si, while the shoots showed a significant decrease in expression. Cl^−^ concentrations in the cytoplasm of leaf mesophyll cells decrease in response to Si. More research is needed on other possible chloride transporters, as well as on Si-imposition in other order of treatments, such as imposing Si to salinity-stressed wheat cultivars (spring and/or fall cultivars).

## Figures and Tables

**Figure 1 plants-11-01162-f001:**
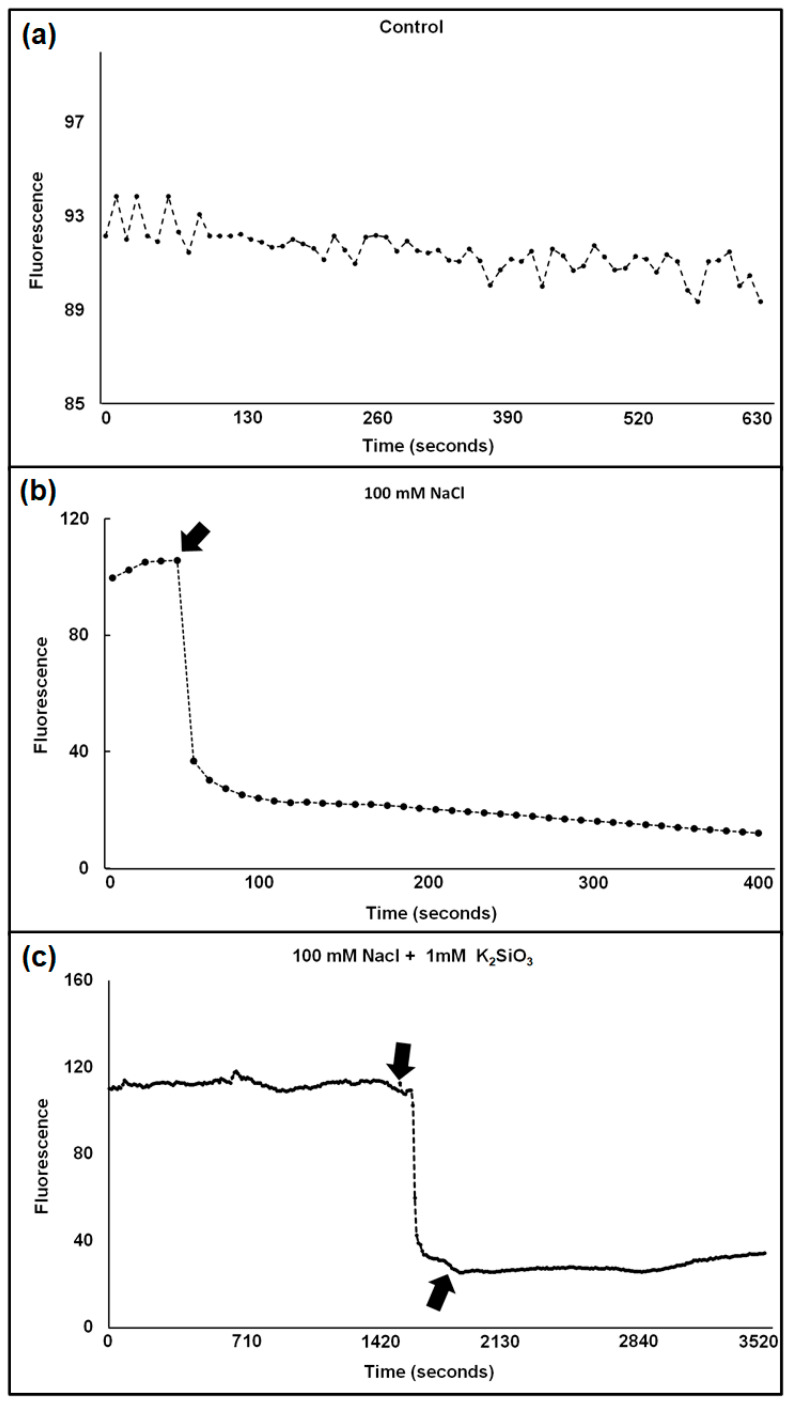
(**a**–**c**) Typical traces show the influx of Cl^−^ into the cytosol of a wheat protoplast in the absence of NaCl (**a**), in the presence of 100 mM NaCl (**b**), and in the presence of 100 mM NaCl + 1 mM K_2_SiO_3_ (**c**). Arrows indicate addition of NaCl (**b**) and of NaCl + 1 mM K_2_SiO_3_ (**c**). Excitation and emission wavelengths were 350 ± 10 and 460 ± 10, respectively.

**Figure 2 plants-11-01162-f002:**
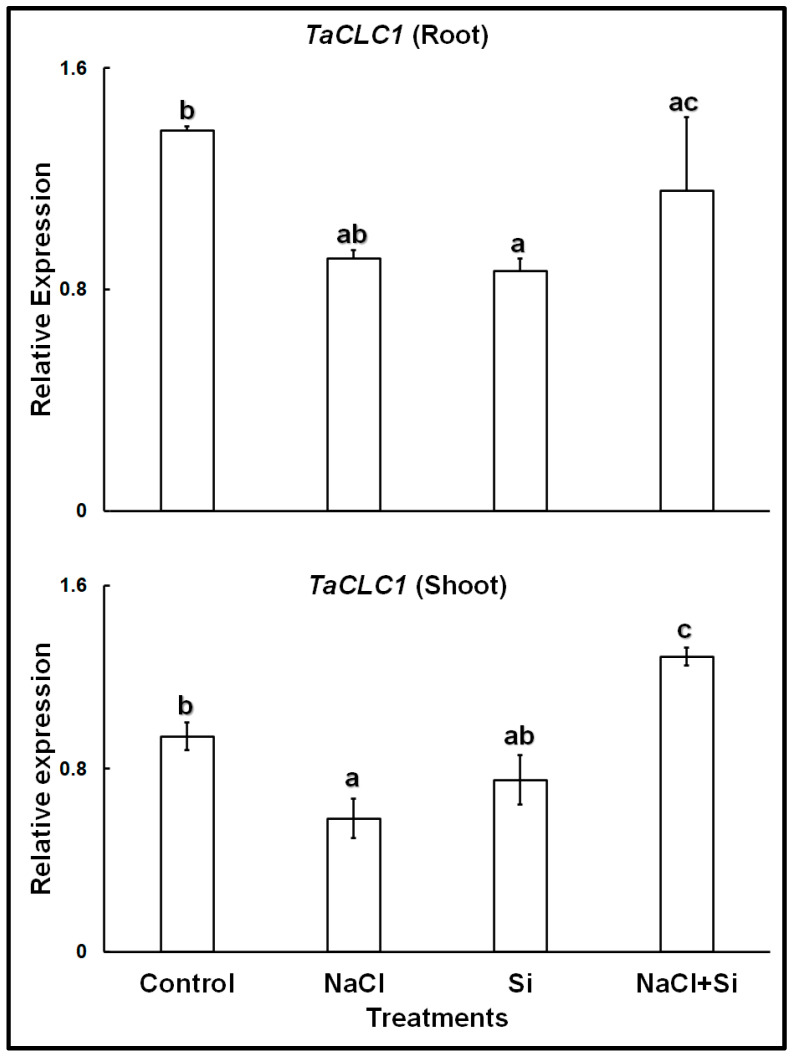
Relative expression of *TaCLC1* in roots and shoots of wheat cv. Vinjette after 4 day of different treatments during the cultivation in nutrient solution. Different letters a–c indicate significant differences (*p* ≤ 0.05) in treatment-dependent expression levels within each organ.

**Figure 3 plants-11-01162-f003:**
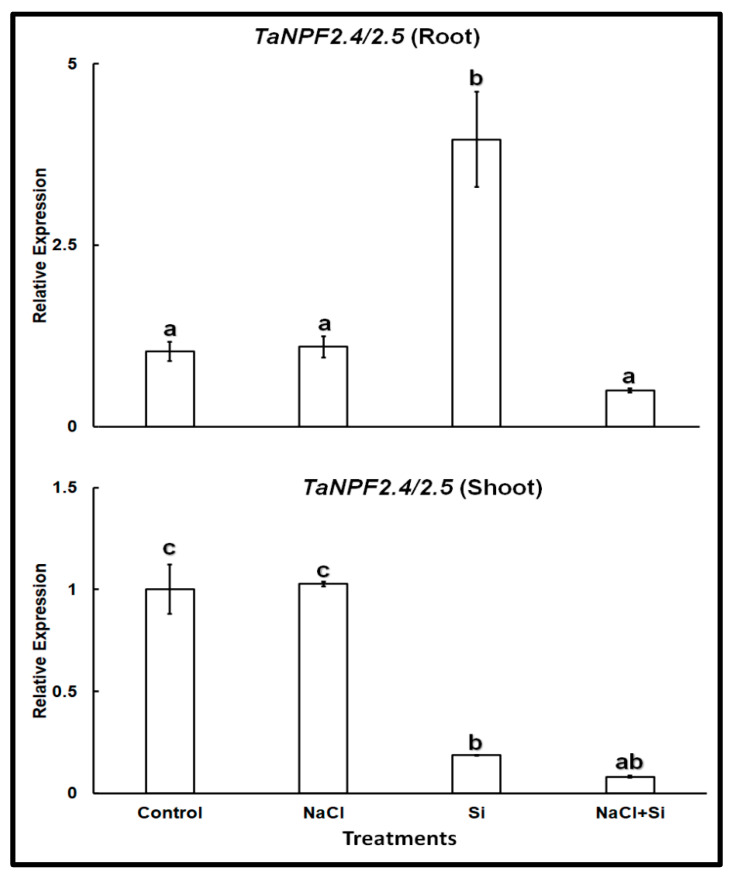
Relative expression of *TaNPF2.4/2.5* in roots and shoots of wheat cv. Vinjette after 4 day of different treatments (100 mM NaCl, 1 mM Si or their combination) during cultivation in nutrient solution. Different letters indicate significant difference at *p* < 0.05 using Tukey’s test.

**Table 1 plants-11-01162-t001:** Fluorescence intensity changes (%) in leaf protoplasts of cv. Vinjett and cv. S-24 loaded with chloride sensitive dye MQAE [N-(6-methoxy-quinolyl) acetoxy ester] upon addition of 1 mM K_2_SiO_3._ Followed by the addition of 50 or 100 mM NaCl. Data are also shown for cases where 50 or 100 mM NaCl addition was followed by the addition of 1 mM K_2_SiO_3_ to the protoplasts of both cultivars. Letters (a–e) indicate significant differences (*p* ≤ 0.05) in fluorescence intensity after different salt treatments with and without K_2_SiO_3._ Letters (y–z) represent significant (*p* ≤ 0.05) difference in fluorescence intensity between cv. Vinjett and cv. S24 receiving the same treatment. A higher fluorescence decrease means a higher chloride concentration. Means ± SE. n = 10.

Treatments		Fluorescence Intensity Decrease (%)	
1st Addition to Protoplast	2nd Addition to Protoplast	cv. Vinjett	cv. S-24
0 mM K_2_SiO_3_	50 mM NaCl	38.21 ± 0.223 cy	33.80 ± 0.202 cz
	100 mM NaCl	50.28 ± 0.305 ay	46.71 ± 0.311 az
1 mM K_2_SiO_3_	50 mM NaCl	30.33 ± 0.212 ey	26.82 ± 0.64 dz
	100 mM NaCl	45.65 ± 0.483 by	43.3 ± 0.36 bz
50 mM NaCl	0 mM K_2_SiO_3_	38.21 ± 0.223 cy	33.80 ± 0.202 cz
	1 mM K_2_SiO_3_	35.97 ± 0.196 dy	33.89 ± 0.163 cz
100 mM NaCl	0 mM K_2_SiO_3_	50.28 ± 0.305 ay	46.71 ± 0.311 az
	1 mM K_2_SiO_3_	47.36 ± 0.397 by	45.99 ± 0.155 az

**Table 2 plants-11-01162-t002:** The gene-specific primers used for RT-qPCR.

Gene	Primers	Sequences	References
*TaCLC1*	Forward	TCGTGGCTGTTGTGGTGCGA	Vicente et al., (2015) [48]
	Reverse	AACCGCCAGCCCCAAAATGACC	
*TaNPF2.4/2.5*	Forward	ACAATGGACTGTCACCTTGGAACAC	Buchner and Hawkesford, (2014) [26]
	Reverse	TGCAGTTAGGGCGATTAA GGATATGG	
*Ta2291*	Forward	GCTCTCCAACAACATTGCCAAC	Paolacci et al., (2009) [49]
	Reverse	GCTTCTGCCTGTCACATACGC	
*Ta2776*	Forward	CGATTCAGAGCAGCGTATTGTTG	Paolacci et al., (2009) [49]
	Reverse	AGTTGGTCGGGTCTCTTCTAAATG	
*Ta54227*	Forward	CAAATACGCCATCAGGGAGAACATC	Mu et al., (2019) [50]
	Reverse	CGCTGCCGAAACCACGAGAC	

## Data Availability

Data are available from Corresponding authors.

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
