# Peer review of "Silicate Inhibits the Cytosolic Influx of Chloride in Protoplasts of Wheat and Affects the Chloride Transporters, TaCLC1 and TaNPF2.4/2.5"

_plants, 2022, doi:10.3390/plants11091162_

Round 1

Reviewer 1 Report

The study that I reviewed examined the effects on two cv of wheat that differ in sensitivity to salt, in the absence and presence of potassium silicate (K2SiO3).

The topic of this article is interesting and current. The article is clearly written, with minimal occurrence of typos. After the required recommendations have been integrated, the manuscript can be published:

Line 31: Add 1. to the subtitle (1.1. Salinity stress in plants)

Line 50: Add a space [3, 15-16]

Line 53: Add 1. to the subtitle (1.2. NPF transporters involved in chloride transport)

Line 68: add 1. to the subtitle, delete the double Tt, add parentheses before and after Triticum aestivum and add L. after aestivum (1.3. NPF transporters in wheat (Triticum aestivum L.))

Line 76: Add 1. to the subtitle (1.4. CLCs channels involved in chloride transport)

Line 77: add L. after aestivum

Line 95: Add 1. to the subtitle (1.5. Silicon effects)

Line 101: Add 1. to the subtitle (1.6. Chloride in the cytosol of plant cells)

Line 123: Add 2. to the subtitle

Line 153: Add 2. to the subtitle

Line 163: In Figure 2 TaCLC1 (Root) the letters of significance are incorrect. I believe that for NaCl it should be ab and not a, that for Si it should be a and not ac and finally that for NaCl + Si it should be ac and not ab

Line 171: In the title of figure 3 add the explanation of the letters of significance

Line 172: In Figure 3 TaNPF2.4 / 2.5 (Shoot) the letters of significance appear to be wrong, perhaps NaCl + Si is a instead of ab?

Line 251: Add 4. in the subtitle (4.1. Plant material and cultivation)

Line 271: Add 4. in the subtitle (4.2. Protoplast isolation and loading with MQAE dye)

Line 280: Add 4. in the subtitle (4.3. Fluorescence measurement)

Line 302: Add 4. in the subtitle (4.4. Statistical analysis)

Line 307: Explain the type of ANOVA used which is probably the one-way ANOVA as indicated in Line 358

Line 310: Add 4. in the subtitle (4.5. Expression analysis of TaCLC1 and Ta NPF2.4 in roots and leaves by RT-qPCR.)

Line 312: Add 4. in the subtitle (4.5.1. Total RNA Extraction)

Line 323: Add 4. in the subtitle (4.5.2. cDNA Synthesis)

Line 329: Add 4. in the subtitle (4.5.3. Real-Time Quantitative (RT-qPCR) analysis)

Line 357-359: Delete this sentence and put it in the statistics section

Reviewer 2 Report

The research manuscript entitled “Silicate inhibits the cytosolic influx of chloride in protoplasts of wheat and affects the chloride transporters, TaCLC1 and TaNPF2.4 /2.5”, from Albert Premkumar and colleagues, focus on the influx of Cl- ions in two different wheat cultivars and in the expression of two Cl- transporters. The manuscript is in general well organized, and the experiments are sound and logical. However, an extensive English revision of the manuscript is needed as several phrase constructions, verbs in the wrong form and misspelling errors were detected. Furthermore, the introduction section must be reformulated: the idea of dividing the introduction into subtopics is good and helps the reader, but some sections must be presented in different orders and will be detailed below. The results section is rather poor, and the figures presented could be more appealing. The discussion and material and methods sections are well presented, with sufficient details and providing enough bibliographic references to support the text.  Finally, the authors mentioned a supplementary figure, but it was not available for evaluation.

Given this, the following list presents several suggestions and/or comments, raised upon a careful reading of the manuscript.

Comments to the manuscript:

  • Line 37 of Introduction - …”and osmoregulation, as activator of enzymes, and as pH.” It is not clear what the authors mean by “and as pH”. Do they mean that Cl- has a role in pH maintenance or regulation? Plea re-phrase.
  • Lines 48-51 of the introduction – The sentence is too long and makes it difficult to understand the message. Could be divided into two. Also, misspelling of the word researchers.
  • Line 55 of introduction – the authors say: “… of the family with 2.5 proteins…”. Is this correct?
  • Point number 3 of the introduction “NPF transporters in wheat….” - The first paragraph should be included in the previous point, as it is the same topic and the second paragraph, where the authors mention the aim of this study, should be moved to the end of the introduction. The information regarding the oligonucleotides for the qPCR should be placed either on the M &M sections or in the results. The information is not needed here and breaks the flow of the introduction.
  • The authors include a topic regarding “silicon effects”, but it is rather disappointing because is too short and the information provided is not sufficient. In fact, the authors mention several (7) bibliographic references, so there must be something to say about the topic. Maybe this would gather more interest if the authors gave more information as, for instance, how does silicon reduce the negative effects of stress? Is there information regarding that topic?
  • Lines 108-113 of the introduction – This paragraph should be moved for the M&M or results sections.
  • This reviewer suggests that the authors provide a new topic, at the end of the introduction summarizing the aims of this manuscript/project, rather than spread them along with the text. The organization and the flow of the manuscript would be greatly improved.
  • Figures – In all figures, the legend must be placed after the figure. The opposite is valid for tables, where the legend appears before the table.
  • In figure 1, the letters a, b and c are missing in the figure as long as the units in the fluorescence axis.
  • Regarding table 1, this reviewer believes that additionally to the table, the authors could present a graphic bar, comparing all the situations under study. It would be easier to understand the message and visually more attractive.
  • Furthermore, images of the fluorescent protoplasts and of the plants used in these treatments could be shown in a figure.
  • Lines 146-149 of results – The sentence is quite confusing. Try to rephrase it.
  • Line 165 of results – “…TaNPF2.4/2.5 showed another expression pattern”. Replace “another” with a “different”.
  • The legend of figure 3 is quite incomplete. The letters a, b, ac, ab, etc... should be mentioned in the legend, along with their meaning.
  • Lines 185-187 of discussion – This sentence is very confusing, and it is difficult to understand what the authors meant. Please rephrase.
  • Line 206 of discussion – the verb is missing in the sentence.
  • Line 210 of discussion – The sentence is confusing. Please rephrase.
  • Line 223 of discussion – There is a reference not well annotated.
  • Line 233 of discussion – Remove “effects of salinity”, and it is already said in the same sentence.
  • Overall, in the discussions section, there are isolated sentences that should be connected and discussed in light of the results presented.
  • Line 257 of M&M – “they” should be replaced by “There”.
  • Line 260 of M&M – The authors mention white polyethene. Is it correct?
  • A section regarding protoplasts preparation from wheat plants is missing in M&M. Must be included.
  • Line 282 of M&M – The authors mention a recovery after centrifugation, but it is not clear what was centrifuged, or when.
  • Figure 4 could be moved to supplementary material.
  • Lines 306-307 of M&M – The sentence “The tables show …” is repeated.
  • Review the italic forms along with the text in the plants' names and in genes/proteins.
  • Some references do not have the doi information. They should be formatted the same way.
  • A PhD thesis is provided as a reference. This is not usual. I am not sure if it is allowed. Please check.

Round 2

Reviewer 2 Report

The research manuscript entitled “Silicate inhibits the cytosolic influx of chloride in protoplasts of wheat and affects the chloride transporters, TaCLC1 and TaNPF2.4 /2.5”, by Albert Premkumar and colleagues, focuses on the influx of Cl- ions in two different wheat cultivars and in the expression of two Cl- transporters.

The revised version provided by the authors addressed most of my initial concerns. The authors did a great job of improving the images presented and correcting the major points referred to. Also, most sections have been re-written and improved, which greatly improved the manuscript and its comprehension. They also provided sound justifications for the questions and doubts that arose during revision and included more data as supplementary figures.